# LiDAR Platform for Acquisition of 3D Plant Phenotyping Database

**DOI:** 10.3390/plants11172199

**Published:** 2022-08-25

**Authors:** Manuel G. Forero, Harold F. Murcia, Dehyro Méndez, Juan Betancourt-Lozano

**Affiliations:** Facultad de Ingeniería, Universidad de Ibagué, Ibagué 730002, Colombia

**Keywords:** 3D maize database, LiDAR platform, plant phenotyping, point clouds, 3D reconstruction

## Abstract

Currently, there are no free databases of 3D point clouds and images for seedling phenotyping. Therefore, this paper describes a platform for seedling scanning using 3D Lidar with which a database was acquired for use in plant phenotyping research. In total, 362 maize seedlings were recorded using an RGB camera and a SICK LMS4121R-13000 laser scanner with angular resolutions of 45° and 0.5° respectively. The scanned plants are diverse, with seedling captures ranging from less than 10 cm to 40 cm, and ranging from 7 to 24 days after planting in different light conditions in an indoor setting. The point clouds were processed to remove noise and imperfections with a mean absolute precision error of 0.03 cm, synchronized with the images, and time-stamped. The database includes the raw and processed data and manually assigned stem and leaf labels. As an example of a database application, a Random Forest classifier was employed to identify seedling parts based on morphological descriptors, with an accuracy of 89.41%.

## 1. Introduction

By 2050, the world’s population is expected to rise by almost 10 billion, while the average rate of increase in crop production is only about 1.3% per year [1]. Therefore, new technologies have been developed to improve agricultural yield without affecting the environment to ensure global food sustainability. One way to improve production is to find among plants those more productive varieties, resistant to diseases or stress, among other advantages. To distinguish them, their phenotype is determined, i.e., the specific observable aspects of a plant or its visible characteristics, such as internal factors, related to the genetics itself, and external factors associated with the environment, adaptation, regulated by abiotic factors [2].

The plant phenotype is formed during plant growth from the influence of the species-specific genotype and its interaction with abiotic and biotic factors. Over the years, phenotyping measurement and analysis have been a laborious, expensive, and time-consuming task, so research in this field has focused on the development of automated, multifunctional and high-throughput phenotyping technologies to advance in breeding [3]. Therefore, in the phenotyping study, the measurement of the 3D morphology of a plant plays an important role. Morphological traits provide a viable way to evaluate stress, yield, growth, anatomy, and overall plant development [4,5,6,7]. Plant morphology can be analyzed at three scales: canopy scale in the field, individual plant and organ scale indoors, and micro-scale in laboratories [8]. Indoor applications usually employ pot-grow plants and combine non-destructive data acquisition techniques such as RGB imaging, depth imaging, and laser scanning.

Currently, single-plant phenotyping platforms are fairly advanced [9,10,11], and their use indoors guarantees adequate light conditions for imaging and low airflow, avoiding disturbances in the measurements. Different studies address population phenotypes by looking for plants that have accelerated growth or better production and thus find better genotypes. Other studies evaluate individual phenotypes, which estimate characteristics such as height, volume, and the number of leaves to obtain parameters to determine the best phenotypes. Among the studies reviewed, none was found that separates the plant organs in order to determine the phenotype, since the particular characteristics of each organ could be used to obtain a better approximation of the phenotype. On the other hand, field platforms for individual plants still need to be improved for more detailed feature extraction [12].

To improve the identification of varieties and their characteristics, the phenotype is analyzed in detail using Machine Learning techniques. For this purpose, it is often important to acquire a large amount of high-resolution and accurate data. Several acquisition techniques have been employed for this purpose, which can be grouped into two classes. The first, employing multiple cameras [13] or one single camera at different angles [14,15,16]. The second uses depth cameras based on Kinect or Light Detection and Ranging (LiDAR) [17,18]. The latter technology has a lower computational cost due to the fewer points on the plant, compared to data obtained using photographic cameras.

According to The Food and Agriculture Organization (FAO), maize is among the world’s five most important crops. It is expected to absorb a significant proportion (more than 22%) of the harvested area by 2050. As a result, more than half of the increase in food demand for cereals is expected to come from maize. Therefore, many researchers have developed efficient, high-throughput phenotyping platforms and methods to acquire traits from maize plants [10,19,20,21,22,23,24]. Although several papers have been published on data acquisition methods for 3D point cloud plant phenotyping using cameras and Kinects [18,25,26], few have been developed for 3D acquisition using LiDAR [27]. Despite these developments, their verification has to be performed with closed and incomplete databases, which presents drawbacks for future research. Therefore, this study presents a new prototype for the acquisition of plant morphological data based on a LiDAR sensor that allows the rotational and translational displacement of the seedling placed on a rotating platform and the vertical movement of the sensor. The platform was tested on a database of indoor potted maize seedlings containing 362 three-dimensional scans and 2749 images.

The structure of the paper is as follows. In Section 2 the development of the LiDAR platform (Section 2.1) and the application of the constructed point cloud (Section 2.2) are discussed. Subsequently, the results obtained in the project are presented and discussed in Section 3. Conclusions are presented in Section 4.

## 2. Materials and Methods

### 2.1. Platform LiDAR

The physical configuration of the developed platform consists of a turntable, driven by a stepper motor, where the plant to be scanned is placed and which allows a 360° rotation at a resolution of 0.1°. As shown in Figure 1, the scanning system consists of a LiDAR sensor (LMS4121R-13000, SICK AG) with visible red light, which emits a laser beam that scans the plant vertically. The main features of the laser sensor are presented in Table 1. The combination of rotary movement and vertical scanning creates a 3D point cloud of the plant with a 360° view.

The LiDAR measurement method is based on the phase correlation principle. The sensor emits a continuous laser beam, which is reflected when it makes contact with an object and is sensed by the scanner’s receiver. The resulting phase delay between the emitted and received 0 is used to determine the distance in centimeters. This device has a scanning frequency of 600 Hz and an aperture angle of 70°. The point furthest from the center of the plant in the horizontal plane is located at a minimum distance of 70 cm from the sensor in the sensor’s working range. This system has some flexibility to set the appropriate LiDAR platform height and distance to the rotation stage depending on the size and shape of the plant.

The LiDAR sensor requires a different power supply than the other devices. For this purpose, an intrinsically safe power supply was used. The communication between the computer system and the LiDAR is also performed through the Transmission Control Protocol (TCP)/Internet Protocol (IP), so a previous configuration of the sensor in the Local Area Network (LAN)/Wide Area Network (WAN) network was necessary.

The sensor was also configured to perform the measurements according to the needs of the proposed system, using the SOPAS ET configuration software. For this purpose, a fixed IP address was initially established for the sensor and a session was initiated to make changes to the predefined parameters in this software. Within the basic configuration, sensor input 1 was set as the control signal and the median and edge detection filters were activated, as this allows better results in data acquisition. Finally, this setting was saved permanently for the continuous use of the sensor, as shown in Figure 2.

For data acquisition, a computer with an Intel Core i7 1.10 GHz × 12 processors and 16 GB RAM was used, running the Ubuntu 18.04 operating system and a Melodic distribution of Robot Operating System ROS. As shown in Figure 3, the stepper motor manipulation was performed by the computer system using an Arduino Nano and a V44 A3967 power driver, synchronized with the laser sensor to acquire a profile of information at each angle of rotation of the platform. Using a high-resolution Logitech Brio camera, the process of taking images of the plants at different angles of rotation was carried out in order to obtain their ground truth information.

The measured distance *S* between the LiDAR and the target, together with the beam angle θ, were obtained using the individual measurement points generated by the sensor. The angle of the rotation stage ϕ was calculated by means of a worm and wheel mechanism adjusted to the resolution of the stepper motor (Figure 4).

The employed motor has a reduction ratio of 1:100, a speed of 3024 RPM, and a number of steps per revolution of 10,000. The worm gear mechanism adds a reduction ratio of 1:36, so the speed is reduced to 0.084 RPM and the number of steps per revolution is increased to 360,000. Hence, the resolution of the complete mechanism is 0.001° and the selected angular resolution of the platform was 0.1°. Since this angle is very small, the vibration of the seedlings is negligible. However, to ensure that it is zero, a rest time of 5 s was given between each acquisition.

Using the coordinate systems illustrated in Figure 4, the vertical scan plane of the LiDAR passing through the center of the rotating disc O is taken as the XZ-plane, having its origin at O. The distance *d* between the LiDAR and the center of the rotation stage, together with the distance *h* and the tilt angle of the sensor φ were calculated by means of a platform calibration process. For this, a target of size 40 cm × 4.5 cm, placed on the disc, centered at O, and made of a low-reflective material to reduce laser beam scattering, was scanned beforehand. The scanning result is a vertical line, from which information about the distances and inclination between the sensor and the turntable is extracted.

The LiDAR measurements were converted into Cartesian XYZ coordinates using homogeneous transformations. Firstly, being a 2D LiDAR, in the X${}^{\prime }$Z${}^{\prime }$ plane, the Cartesian coordinates of the sensor are defined by Equation (Equation 1).
(1)$$\left[\begin{array}{c} X^{\prime }\\ Y^{\prime }\\ Z^{\prime }\\ \omega^{\prime } \end{array}\right]=\left[\begin{array}{c} -s*cos(\theta )\\ 0\\ s*sin(\theta )\\ 1 \end{array}\right].$$

This plane must then be rotated around the Y${}^{\prime }$ axis, taking into account the tilt of the sensor in the X${}^{\prime }$Z${}^{\prime }$ plane. For this purpose, the following homogeneous transformation is used:(2)$$R=\left[\begin{array}{cccc} 1 & 0 & 0 & 0\\ 0 & cos(\varphi ) & -sin(\varphi ) & 0\\ 0 & sin(\varphi ) & cos(\varphi ) & 0\\ 0 & 0 & 0 & 1 \end{array}\right]\left[\begin{array}{c} X^{\prime }\\ Y^{\prime }\\ Z^{\prime }\\ \omega^{\prime } \end{array}\right].$$

Then, to align the coordinate planes, the following translation transformation is used.
(3)$$t=\left[\begin{array}{cccc} 1 & 0 & 0 & d\\ 0 & 1 & 0 & 0\\ 0 & 0 & 0 & h\\ 0 & 0 & 0 & 1 \end{array}\right]*R.$$

Finally, to obtain the XYZ coordinates of each LiDAR scan within the reference frame, a rotation transformation is performed given by the disk angle ϕ, as shown in Equation (Equation 4). Algorithm 1 describes the process of acquisition and reconstruction of the 3D point cloud using the notation presented in Figure 4. It includes the most relevant steps, such as platform calibration, sensor data acquisition, the transformation of the data into homogeneous coordinates, and the generation of the point cloud and images.
(4)$$\left[\begin{array}{c} X\\ Y\\ Z\\ \omega \end{array}\right]=\left[\begin{array}{cccc} cos(\phi ) & -sin(\phi ) & 0 & 0\\ sin(\phi ) & cos(\phi ) & 0 & 0\\ 0 & 0 & 1 & 0\\ 0 & 0 & 0 & 1 \end{array}\right]*t.$$

The platform control was written in Python 3.6, using the serial, opencv, cv_bridge, and open3d libraries to process the information obtained from the devices. The distance and intensity measurements sensed by the LiDAR were obtained with the ros node sick_lms_4xxx [28] and the images were acquired with the Logitech camera using the usb_cam node [29]. A camera calibration process was previously performed with the ros package camera_calibration using a checkerboard as a target, obtaining the camera, distortion, rectification, and projection matrices. Using the ros package image_proc the image distortion was removed, which was registered in the ros topic image_rect_color, as shown in Figure 5. Finally, through the multiplatform framework kivyMD and its respective libraries and ros nodes (simple_gui), an interface between the different device control commands and the user was created.

The graphic interface shown in Figure 6 was implemented in order to allow interaction between the user and the prototype. The interface executes in a specific order the different rosnodes used for the initialization of the platform devices and data collection. Then, a calibration process is carried out, which must be repeated each time the platform where the plant is located is moved. In addition, it allows the input of the parameters for the experiments, such as the angular resolution of the imaging platform θ, initial angle ($\omega_{i}$), and final angle ($\omega_{f}$) of scanning. As a result of the 3D scan, three types of files have generated A file in TXT flat format with the information of the Cartesian coordinates of the 3D model obtained; a set of RGB images taken throughout the process and another set of the same size in rosbag format with the synchronized information of the rostopics used during the 3D reconstruction process.

To determine the accuracy of the measurements at each coordinate, a small cube was scanned, since only the length of one of its edges needs to be detected to know its size. In this way, the lengths of the cube’s sides were obtained, as shown in Figure 7. The edges located on the top face of the cube were denoted with the letter “U”, the front ones with “L”, and those on the bottom face with “D”.

**Algorithm 1:** 3D model generation with the proposed LiDAR scanning platform.

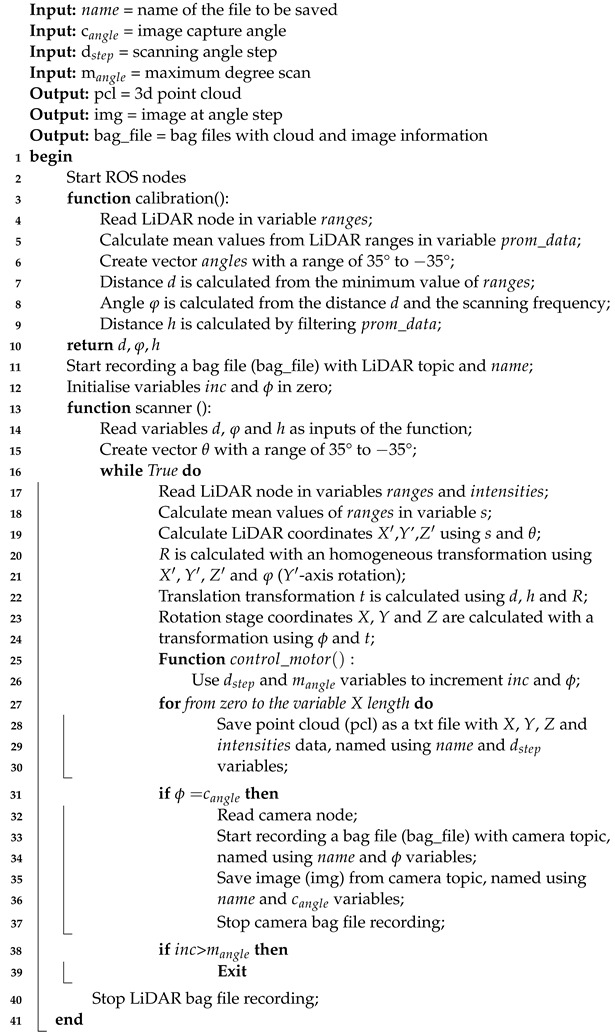



Once the edge lengths were obtained, the measurement error was calculated as follows:(5)$$\mathrm{Indiv}.\;\mathrm{Error}\left[\%\right]=\left|\frac{m-Ref}{Ref}\right|*100,$$
where *m* is the measured value and Ref is the actual one. The measurement errors in the X, Y, and Z coordinates were obtained as the average of the individual errors per axis. The average accuracy is given by the mean of the errors in each coordinate and the absolute error of all measurements by:(6)$$\mathrm{Abs}.\;\mathrm{Error}=\sqrt{\frac{\sum {\left(m-\bar{m}\right)}^{2}}{N(N-1)}},$$
where $\bar{m}$ is the mean value and *N* is the amount of data.

To determine the accuracy of the measurements made with the platform, 142 repetitions of the distance determined from the point cloud were compared with a reference measurement obtained manually. The CloudCompare software was used to determine the distance in the point cloud, taking into account that the measurements obtained with the LiDAR correspond to the light beam emitted with a sweep angle of zero degrees with respect to the horizon, which corresponds to the case with the minimum angle of incidence. This process was repeated four times varying the distance between the sensor and the cube, in a range between the minimum value detected by the LiDAR and the maximum length allowed by the platform, as shown in Figure 8. Once the experiment was performed, a mean dispersion of 0.071 cm was obtained. Thus, a third-degree polynomial regression was performed, resulting in Equation (Equation 7), which represents the trend of the precision of the sigma platform as a function of the distance between the seed and the *d* laser sensor.
(7)$$\sigma_{\left(d\right)}=0.0639d^{3}+0.1139d^{2}+0.0473d+0.0589.$$

It is worth noting that the accuracy measured using a light beam at an angle of incidence close to zero degrees and that, although not evident in the above equation, the scattering is proportional to the angle of incidence between the light beam and the illuminated object, which has a significant impact on the quality of the three-dimensional reconstruction and, consequently, affects the determination of phenotypic characteristics. For this reason, each profile acquisition performed by the platform is performed one hundred times to obtain the average value of each point at each angle and to accumulate a measurement with measurement noise reduction in the final reconstruction.

A database of maize plants at different phenological stages was constructed using the proposed platform by obtaining 362-point clouds. For this purpose, 38 seeds were planted in pots, and days after planting (DAP) were used as a parameter. The cultivated maize plants were scanned under laboratory conditions by placing the pots on the platform and measuring the light intensity in lux at the initial instant. The process was carried out with a first sowing up to the scion stage and six more sowings up to the seedling stage, to evaluate different growth scenarios.

The angle of incidence, light conditions, material, texture, and measuring range influence the fidelity of the measurements and thus the exact determination of the phenotypic characteristics acquired from the point clouds. Some of these factors affect the quality of the measurements, producing noise. Thus, the reflectance of the material, its color, and its texture cause the laser to deflect and produce unwanted values. Noise can be removed by using classical filtering techniques such as Statistical Outlier Removal [30].

Due to the nature of the laser, when the structure to be reconstructed is very thin, the beam passes through it and is not detected. For this reason, some parts of the seedling will have disconnected sections. To solve this problem, different techniques based on mathematical morphology or growth by regions are used. Another type of solution consists of reconstructing the missing sections from the information obtained from photographs taken of the seedling simultaneously during the acquisition of the point cloud.

### 2.2. Application

To demonstrate the reconstruction reliability, a phenotypic analysis of characteristics such as height, volume, and classification of seedling organs was performed. For such phenotypes, it is first necessary to segment the point cloud. This is performed by limiting the working area on the Z-axis, eliminating the pot and outliers above the maximum allowed height of the seedling.

Seedling height *h*:To determine the height of the plant, the point in the cloud with the highest value on the Z-axis is required.
(8)$$h=max_{z}.$$Total Volume *v*:To calculate the volume of the plant, a voxelization of the points with a distance of 25 cm is performed. Then the voxel count is denoted in the (Equation 9) equation as the summation of the V parameter is performed and multiplied by the distance value used. The distance value was calculated experimentally.
(9)$$v=2.5\times 10^{-3}\sum_{i=1}^{N}V_{i}.$$Classification of organs:In order to separate the organs of the seedling, a classification is made with respect to the stem and leaves. The database is split in two, taking the first days of shoot and seedling up to the third leaf as one database and the rest as another. In each database, 60% of the point clouds are used for training a Random Forest classifier, 20% for tuning the classifier parameters, and the rest for model validation.Once this result has been obtained, filtering of leaf segments that were considered stems is carried out. To do this, a virtual ring is used that goes up from the base of the plant to the highest point [31]. The radius of the ring was 0.0015 and was estimated and validated experimentally.

## 3. Results and Discussion

As mentioned in Section 2, the point cloud of a cube-shaped reference object was used to establish the precision error of the reconstruction performed. The results obtained are shown in Table 2. As can be seen, the error was less than 3 cm with respect to the ground truth.

The constructed platform was used for 3D scanning of plants at different phenological stages. Figure 9a shows the development of a plant from the first plantation, which was scanned during three stages of its life cycle. The growth process of a plant from the second plantation is also presented (see Figure 9b). The color scale in the point clouds corresponds to the intensity values delivered by the LiDAR sensor. Each reconstruction took about 42.5 min with the default values set in Section 2.

A database containing 362 seedlings was created using the designed platform. In Table 3 it can be seen each group of data taken from each planting campaign. The database, which will be freely accessible, is distributed in seven folders, one per campaign. Each one contains the folders of each plant, with an identifying name and the day of scanning according to its DBH. Each folder includes eight images of the plant taken every 45 degrees, a file with the raw point cloud taken with the LiDAR, and a file containing the processed point cloud, which contains the stem and leaf labels.

As mentioned in Section 2.2, the database was used to estimate the phenotypic characteristics of each plant. Figure 10 shows the percentage error obtained by comparing the actual height of eight randomly selected plants with the estimated one. Table 4 presents the estimated volume of a seedling over time and Table 5 the stem and leaf classification accuracy obtained with 70 samples, which was on average 89.41%.

## 4. Conclusions

This paper presents a new prototype LiDAR platform for plant phenotyping in a controlled environment. The acquisition time is relatively long (42.5 min), in order to obtain a high accuracy (0.0325) and precision (0.0310) of the points obtained. With this equipment, a freely accessible database of 362 maize plants was constructed and used to obtain three phenotypic parameters, height, volume, and classification of stalks and leaves, in order to verify the reliability of the database. The first two parameters were compared with the real values, obtaining an error of 2.3% and 7.1% respectively.

## Figures and Tables

**Figure 1 plants-11-02199-f001:**
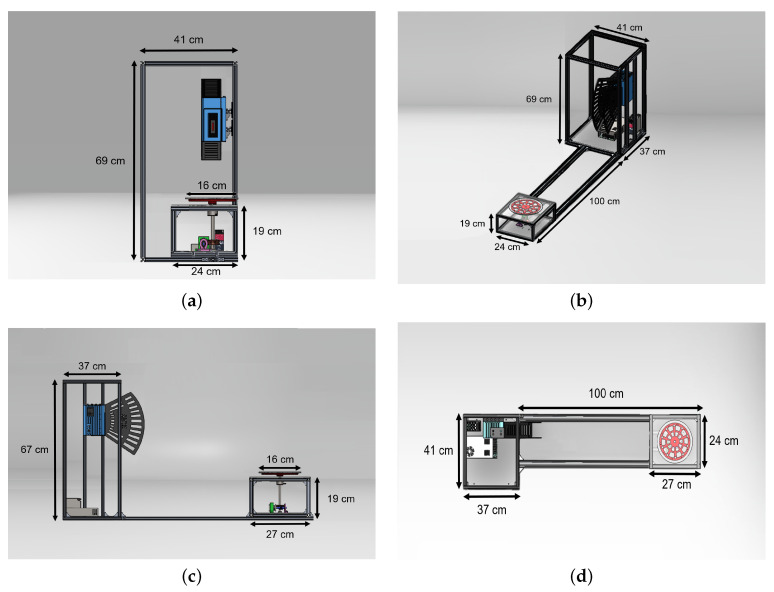
Illustration of the proposed platform based on a LiDAR sensor and a turntable for 3D plant reconstruction. (**a**) Frontal view, (**b**) Isometric view, (**c**) Top view, (**d**) Lateral view.

**Figure 2 plants-11-02199-f002:**
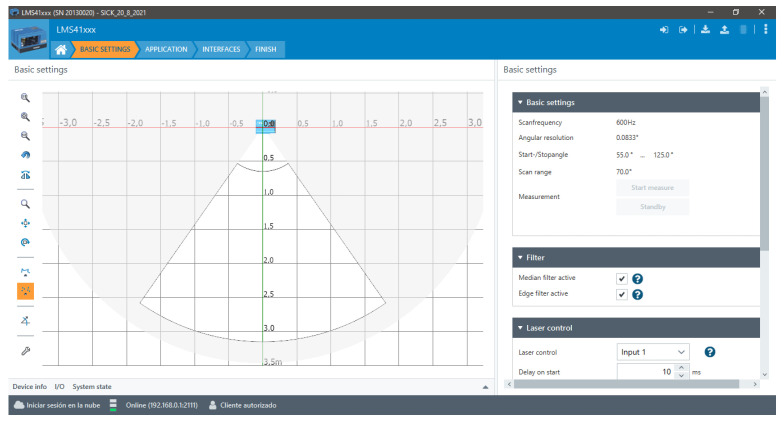
LiDAR configuration in the SOPAS ET software.

**Figure 3 plants-11-02199-f003:**
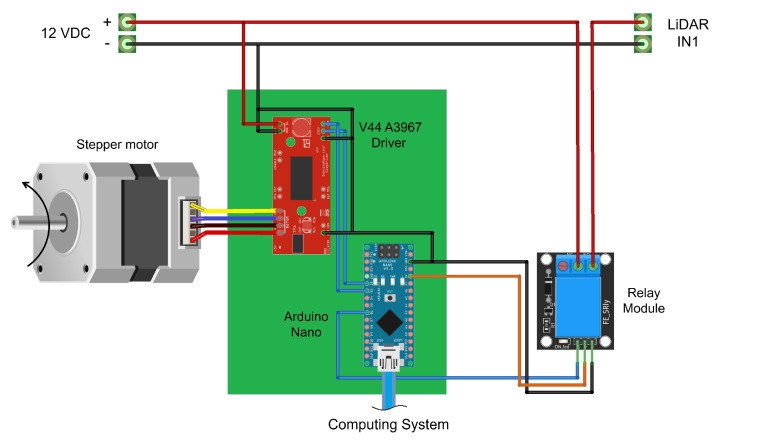
Schematic diagram of the connections for the electronic devices.

**Figure 4 plants-11-02199-f004:**
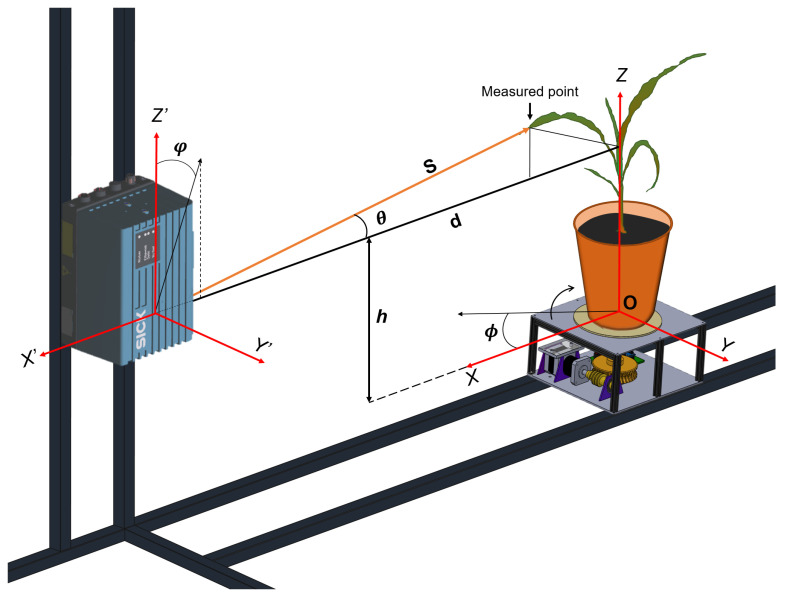
Schematic of the developed scanning system. The functional diagram shows the LiDAR device and the rotation platform and its reference frames.

**Figure 5 plants-11-02199-f005:**
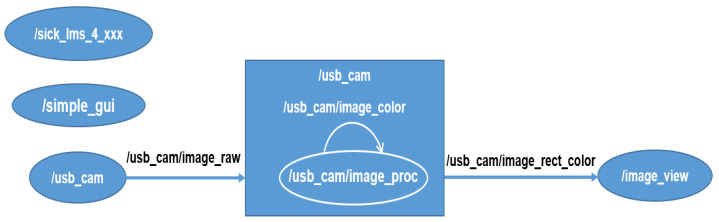
Diagram of the active ros nodes during 3D reconstruction.

**Figure 6 plants-11-02199-f006:**
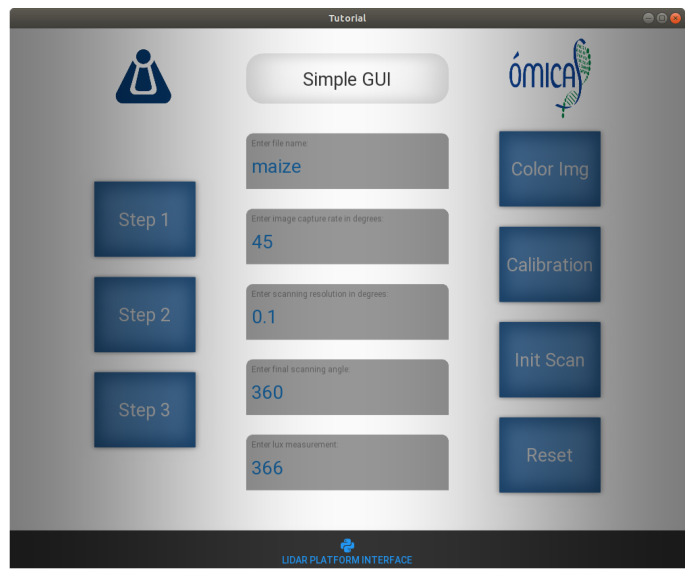
User Interface for LiDAR platform.

**Figure 7 plants-11-02199-f007:**
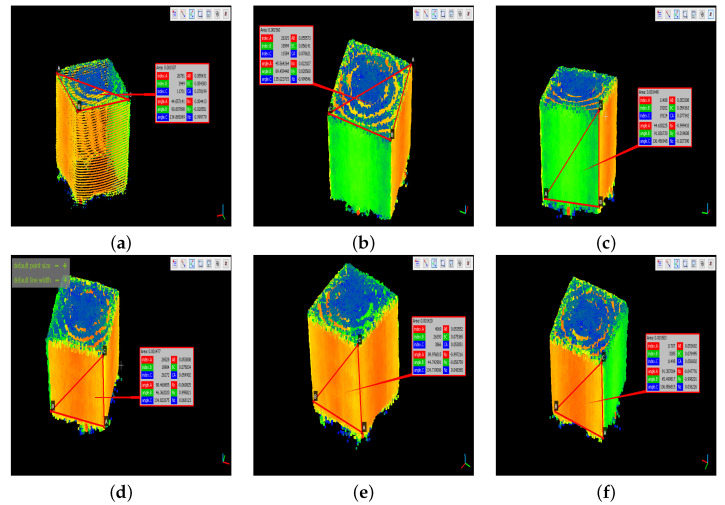
Measurement of the cube edges. (**a**) U1 and U2. (**b**) U3 and U4. (**c**) L1 and D1. (**d**) L2 and D2. (**e**) L3 and D3. (**f**) L4 and D4.

**Figure 8 plants-11-02199-f008:**
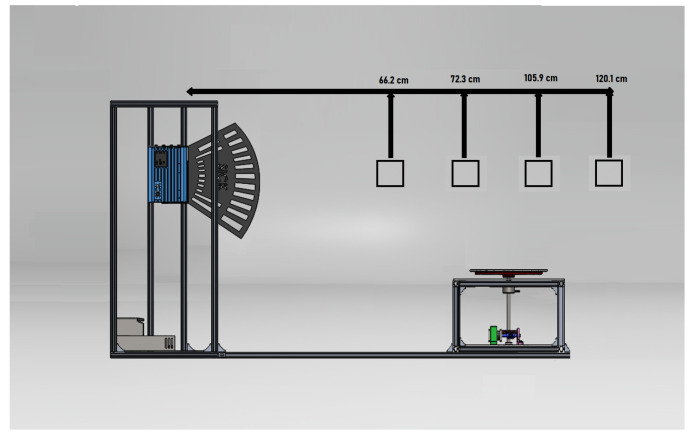
Diagram of the lengths used to measure the precision were measured with the proposed device.

**Figure 9 plants-11-02199-f009:**
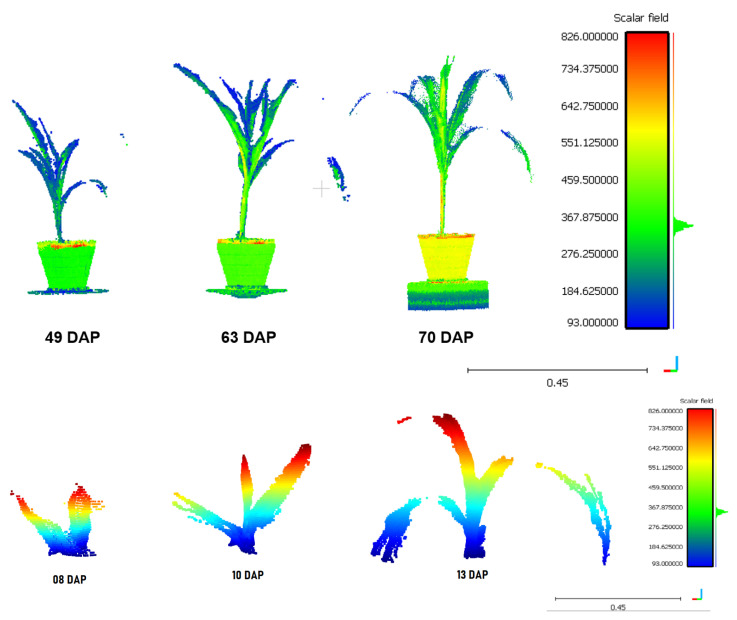
Three-dimensional (3D) point clouds of plants 01 and 08 in three growth phases. The color scale corresponds to the signal strength received by the Lidar.

**Figure 10 plants-11-02199-f010:**
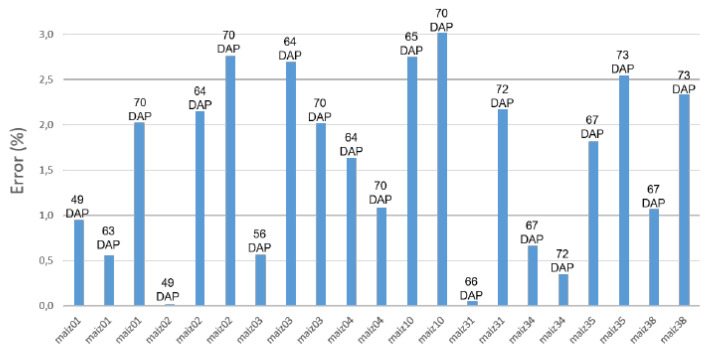
Height estimation error of 8-point clouds.

**Table 1 plants-11-02199-t001:** Main features of LiDAR sensor.

Feature	LMS4121R-13000
Application	Indoor
Reading field	Front
Light source	Visible red light
Laser class	2 (IEC 60825-1:2014, EN 60825-1:2014)
Aperture angle	70°
Scanning frequency	600 Hz
Angular resolution	0.0833°
Working range	70 cm … 300 cm

**Table 2 plants-11-02199-t002:** Measurement error obtained with the cube’s 3D point cloud.

Ground Truth			Point Cloud		Error Calculation		
**Coord.**	**ID**	Ref **[cm]**	m **[cm]**	**Indiv**. **Error [%]**	**Prom.** **Error [%]**	$m-\bar{m}$	${\left(m-\bar{m}\right)}^{2}$
X	U1	5.5	5.4383	1.1218	1.6773	0.0134	0.0002
	U2	5.5	5.5573	1.0418		0.1056	0.0111
	D1	5.5	5.3808	2.1673		0.0709	0.0050
	D2	5.5	5.3692	2.3782		0.0825	0.0068
Y	U3	5.5	5.5431	0.7836	2.1418	0.0914	0.0084
	U4	5.5	5.6141	2.0745		0.1624	0.0264
	D3	5.5	5.3308	3.0764		0.1209	0.0146
	D4	5.5	5.3552	2.6327		0.0965	0.0093
Z	L1	5.5	5.4363	1.1582	1.6755	0.0154	0.0002
	L2	5.5	5.4902	0.1782		0.0385	0.0015
	L3	5.5	5.3051	3.5436		0.1466	0.0215
	L4	5.5	5.6002	1.8218		0.1485	0.0220
Accuracy [%]					1.8315		
$\bar{m}$			5.4517			$\Sigma {\left(m-\bar{m}\right)}^{2}$	0.1271
Abs. Error [cm]							0.0310

**Table 3 plants-11-02199-t003:** Point clouds of maize plants separated by campaigns with their respective acquisition dates and number of scanned data.

Campaign	Date	# Plants	Link
First	7 July 2021 to 21 October 2021	21	https://osf.io/fcgwk/
Second	21 October 2021 to 12 November 2021	40	https://osf.io/x5cn9/
Third	2 February 2022 to 18 February 2022	45	https://osf.io/5vykw/
Fourth	28 February 2022 to 23 March 2022	80	https://osf.io/ks7my/
Fifth	28 March 2022 to 9 April 2022	55	https://osf.io/tnhxy/
Sixth	25 April 2022 to 17 May 2022	80	https://osf.io/h63jq/
Seventh	23 May 2022 to 14 June 2022	41	https://osf.io/7uvm4/

**Table 4 plants-11-02199-t004:** Estimation of the volume of a seedling in its different temporal stages.

Maiz08	
**TAP (h)**	**Volume (cm${}^{3}$)**
147.75	1.2068
167.91	1.5688
176.96	3.6859
192.02	7.3712
200.48	11.5586
297.30	19.7802
321.05	24.1338
345.16	38.5767
384.02	67.4274
432.32	100.5300
456.84	107.0390

**Table 5 plants-11-02199-t005:** Accuracy obtained on 70 specimens by classifying their stems and leaves respectively.

Name (Campaign 6)	Accuracy (%)	Name (Campaign 6)	Accuracy (%)
01_01	93.44	05_11	93.34
01_02	89.16	06_01	98.92
01_03	97.97	06_02	87.41
01_04	89.03	06_03	94.10
01_05	78.71	06_04	87.31
01_06	93.69	06_05	98.02
01_07	83.61	06_06	86.13
01_08	91.26	06_07	89.54
01_09	95.12	06_08	82.57
01_10	92.23	06_09	78.65
01_11	91.63	06_10	79.32
02_04	90.34	06_11	91.80
02_05	88.81	09_01	90.91
02_06	86.30	09_02	96.38
04_01	95.75	09_03	95.66
04_02	94.59	09_04	90.42
04_03	91.96	09_05	93.55
04_04	96.02	09_06	94.53
04_05	84.09	09_07	89.51
04_06	97.00	09_08	94.05
04_07	95.52	09_09	95.33
04_08	89.24	09_10	90.12
04_09	98.79	09_11	94.18
04_10	92.25	09_12	94.64
04_11	90.64	12_01	90.00
04_12	87.11	12_02	87.63
05_01	79.91	12_03	81.86
05_02	88.58	12_04	90.02
05_03	90.22	12_05	89.71
05_04	91.42	12_06	91.28
05_05	93.28	12_07	86.62
05_06	95.22	12_08	85.76
05_07	86.77	12_09	90.09
05_08	94.34	12_10	90.77
05_10	96.21	12_11	96.06
Average accuracy = 89.41%			

## Data Availability

Our generated dataset is available online at: 1st campaign: https://osf.io/fcgwk/; 2nd campaign: https://osf.io/x5cn9/; 3rd campaign: https://osf.io/5vykw/; 4th campaign: https://osf.io/ks7my/; 5th campaign: https://osf.io/tnhxy/; 6th campaign: https://osf.io/h63jq/; 7th campaign: https://osf.io/7uvm4/.

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
