# Peer review of "LiDAR Platform for Acquisition of 3D Plant Phenotyping Database"

_plants, 2022, doi:10.3390/plants11172199_

Round 1

Reviewer 1 Report

The paper describes a LiDAR-based phenotyping platform and a database acquired by the prototype system. The authors described the sensor configurations in detail, including a diagram of the connections for the sensors, control and processing units. In addition, the authors provided an overview of the data acquisition process and further evaluated the measurement error on a reference object. These information covers the design, data acquisition and quality evaluation of the proposed prototype and provides valuable references for the plant phenotyping community to replicate their design.

The main question that arises comes from the fact that the system is based on a 2-D scanner and a turntable. How do the authors control the motion of the plant or leaves as the table turns? Especially at more mature growth stages?

The authors estimate the ranging accuracy as a function of range. However, observing the point cloud one immediately observes noisy points around the plant surface, e.g., Fig. 9. Their origin should be explained, especially as such point have greater potential to affect classification algorithms, more than the ranging accuracy. Also, it appears that there are missing data parts (same Figure). What is the reason for that? How can this be resolved?

Some minor comments:

1.      The authors evaluated the precision of the platform as a function of the distance between the plant and the laser scanner. The units for these variables should be specified for better understanding.

2.      The authors demonstrated the application for organ classification using the collected dataset. The motivation of this experiment is not well-explained, and it might be confusing as the readers do not know what to anticipate.

3.      Abbreviations are not spelled out when first presented.

4.      The links to the references in the paper are missing not allowing to judge the review in the paper.

5.      The review of other 3-D reconstruction approaches is very limited, giving the impression that LiDAR-based solutions is essentially the only way. As the paper is about data acquisition and a bit about subsequent processing, this should be expanded.

Reviewer 2 Report

The article “LiDAR Platform for Acquisition of 3D Plant Phenotyping Database” has the objective to describe a platform for seedling scanning using 3D Lidar with a database acquired for use in plant phenotyping research. The subject of the article is important for providing information for the plant phenotyping studies. However the paper is incomplete and then is not ready to be considered for publication. The references in the text and the list of them are missing in the manuscript. The English form needs to be improved to be clear for the readers.

Some specific comments

Title- Adquisition  ---  Acquisition

References (?)

L.66 - Conclusions and future work are presented in Section 4.  ---  No future work was presented.

L.107 - (Figura 4).  ---  (Figure 4).

L.157- To determine the accuracy error  ---   To determine the accuracy

Figure 8 - to measure the precision error  --- to measure the precision

Round 2

Reviewer 2 Report

The article “LiDAR Platform for Acquisition of 3D Plant Phenotyping Database” has the objective to describe a platform for seedling scanning using 3D Lidar with a database acquired for use in plant phenotyping research. The subject of the article is important for providing information for the plant phenotyping studies. In the revised manuscript, the paper was completed following the suggestions of the reviewers. The references in the text and the list of them are included in the revised manuscript. Then the manuscript has been improved by the authors according to the reviewers comments and suggestions.

 Some specific comments

L.15- redTherefore, ?